ecology/environmental science

coral reef, bleaching, parrotfish, *Halimeda*, sediment

**Author for correspondence:**
Chris T. Perry
e-mail: c.perry@exeter.ac.uk

# Bleaching-driven reef community shifts drive pulses of increased reef sediment generation

Chris T. Perry[1], Kyle M. Morgan[2], Ines D. Lange[1] and Robert T. Yarlett[1]

[1]Geography, College of Life and Environmental Sciences, University of Exeter, Exeter EX4 4RJ, UK
[2]Asian School of the Environment, Nanyang Technological University, Singapore 639798, Singapore

  CTP, 0000-0001-9398-2418; KMM, 0000-0002-3412-703X

The ecological impacts of coral bleaching on reef communities are well documented, but resultant impacts upon reef-derived sediment supply are poorly quantified. This is an important knowledge gap because these biogenic sediments underpin shoreline and reef island maintenance. Here, we explore the impacts of the 2016 bleaching event on sediment generation by two dominant sediment producers (parrotfish and *Halimeda* spp.) on southern Maldivian reefs. Our data identifies two pulses of increased sediment generation in the 3 years since bleaching. The first occurred within approximately six months after bleaching as parrotfish biomass and resultant erosion rates increased, probably in response to enhanced food availability. The second pulse occurred 1 to 3 years post-bleaching, after further increases in parrotfish biomass and a major (approx. fourfold) increase in *Halimeda* spp. abundance. Total estimated sediment generation from these two producers increased from approximately 0.5 kg $CaCO_3$ m$^{-2}$ yr$^{-1}$ (pre-bleaching; 2016) to approximately 3.7 kg $CaCO_3$ m$^{-2}$ yr$^{-1}$ (post-bleaching; 2019), highlighting the strong links between reef ecology and sediment generation. However, the relevance of this sediment for shoreline maintenance probably diverges with each producer group, with parrotfish-derived sediment a more appropriate size fraction to potentially contribute to local island shorelines.

## 1. Introduction

The global coral bleaching event of 2014–2017 had widespread impacts on coral reefs [1]. These impacts included extensive coral mortality over large spatial scales [2] and at some sites to depths of in excess of 20 m [3]; associated shifts in coral community assemblages resulting from selective coral mortality

[4] and alterations to reef-associated fish communities [5,6]. The net effects of bleaching have thus been significant, leading to changes in abundance of species that contribute to reef ecological structure and diversity. Many of these impacted species also contribute to the production, erosion or cycling of calcium carbonate on reefs, and thus bleaching events can also modify rates of reef carbonate production and accumulation and reef-derived sediment supply. A number of studies have documented the marked declines that occurred in coral carbonate production rates after the 2014–2017 bleaching event [7,8], and which will affect, at least in the short-term, reef growth potential [9]. By contrast, quantitative data on how disturbance events impact rates of biologically driven reef sediment generation are absent.

This knowledge gap has significant implications for ecosystem and coastal management because of the critical importance of reef-derived sediment for sustaining shorelines and reef islands [10], especially in localities where sediment generation is dominated by a restricted suite of taxa [11]. Reefal sediments derive largely from the following sources: (i) the biological erosion (or bioerosion) of reef framework, most commonly by specific species of fish (especially parrotfish), sea urchins and endolithic taxa (including excavating sponges) [12–14]; (ii) from direct inputs by carbonate secreting taxa, such as molluscs and foraminifera [15], and (iii) through the growth and breakdown of calcareous red (e.g. crustose coralline algae) and green algae (e.g. *Halimeda* spp.) [16,17]. Intuitively, it is reasonable to assume that any major changes in reef ecology, in response to either direct human disturbances or climate-driven stressors, will influence sediment generation rates and the proportions of reef-derived sediment constituents [18]. Quantitative data on how disturbance events impact on reef-scale sediment production are, however, sparse, but what information we do have suggests that marked changes may indeed occur as the composition of reef communities are altered.

There is, for example, clear evidence for both long-term and selective overfishing reducing parrotfish abundance and erosion rates [11,19], with evident implications for reef carbonate budgets [20]. A by-product of parrotfish erosion is the semi-continuous excretion of sand-grade carbonate [13] and, therefore, reduced parrotfish biomass/erosion may also alter the rates of sediment generation on overfished reefs. However, different functional groups of parrotfish are responsible for variable amounts of bioerosion, with higher rates associated with 'excavator' species and significantly lower rates with 'scraper' species [21]. The nature of parrotfish assemblage change (e.g. species, size, functional group) will thus differentially modify sediment generation. More widely in terms of changes in the abundance/composition of carbonate producers (i.e. the source organisms), there is: (i) evidence for local nutrient enrichment changing benthic reef foraminifera populations [22], which are a major shoreline sediment constituent in many parts of the Pacific [23]; (ii) for point-source terrigenous sediment inputs changing the composition of proximal reef sediments [24]; and (iii) for reef tract-scale benthic community shifts changing reef sediment compositions in the Caribbean [25]. These studies support existing conceptual ideas about how ecological disturbances drive changes in sediment generation [18]. However, while it is clear that different types of disturbance may have divergent reef sediment generation outcomes, quantitative data are needed to fully understand the implications for carbonate sediment-dominated landforms. Here we investigate these issues by quantifying the magnitude of change in sediment generation following the 2016 coral bleaching event across reef sites in the southern Maldives, specifically for two locally important sediment-generating taxa, the parrotfish and the calcareous green algae *Halimeda* spp. We quantify changes in the total amounts and the proportions of different sediment size fractions generated over a 3-year period (Jan 2016 to Jan 2019), consider the drivers of these changes, and the potential implications for reef to island sediment supply regimes.

# 2. Material and methods

## 2.1. Study site and ecological data collection

Ecological data were collected in January and September 2016, March 2017 and January 2019, from the southwestern margins of five uninhabited atoll interior reefs in Gaafu Dhaalu atoll, southern Maldives. All data were collected from sites along the outer reef flat/upper reef front (approx. 2 m depth contour) (electronic supplementary material, figure S1). These reefs are minimally influenced by local disturbances, i.e. fishing pressure is low, and immediate point sources of nutrient input are absent. However, reefs were impacted by the strong ocean warming event that started in late March 2016 and persisted at levels above regional bleaching thresholds (approx. 30.9°C) until mid-May 2016, resulting

in widespread bleaching-induced coral mortality [8]. Benthic community composition at each site was quantified using data collected along five replicate transects (each 10 m) at each site using the ReefBudget methodology [26]. Recorded groups included scleractinian corals to the genera and morphological level, e.g. *Acropora* branching, *Porites* massive, etc.; crustose coralline algae (CCA); turf algae; fleshy macroalgae; *Halimeda* spp.; sediment; rubble and other benthic organisms. All data were collected as a function of the true three-dimensional surface of the reefs, thus including cover on overhangs and vertical surfaces, and are reported as mean $\% \pm$ s.d. Parrotfish abundance (ind. ha$^{-1}$) was quantified via underwater visual census (UVC) along eight $30 \times 4$ m belt transects in the same region of each reef in each time period, with all surveys completed by the same experienced observer (K.M.M.). Details on parrotfish species, life phase (juvenile, initial and terminal) and total length (in size classes of 10 cm) were recorded for each individual. Parrotfish biomass (kg ha$^{-1}$) for each species and size class was then calculated using established length–weight relationships and multiplied by fish abundance, following the approach described in [27].

## 2.2. Parrotfish and *Halimeda* spp. sediment generation

Parrotfish bioerosion rates (kg CaCO$_3$ m$^{-2}$ yr$^{-1}$) were calculated for each individual greater than 10 cm using a model based on species, body length and life phase (as reported in [27]). The model was updated with bite rate and scar size data for the dominant central Indian Ocean parrotfish species [28]. Bioerosion rates of parrotfish equate to sediment generation rates because it is generally assumed that all ingested bioeroded substrate is excreted as sediment [13], and there is no evidence of intestinal dissolution of ingested bioeroded reef substrate occurring [29]. Therefore, bioerosion and sediment generation are considered equivalent and are discussed as such here.

To estimate sediment production rates (kg CaCO$_3$ m$^{-2}$ yr$^{-1}$) by *Halimeda* spp. at each site and time point, we used data on the rates of production calculated at these same sites in Jan and Sept 2016 for the two dominant species: *H. macrophysa* and *H. micronesica* [30]. These rates were based on plant dimension metrics so as to quantify plant volumetric space (cm$^3$), from which an empirical relationship between individual plant volume and carbonate production rate was calculated. These data suggest that *H. macrophysa* and *H. micronesica* produce on average 11.8 and 10.9 g, respectively, of CaCO$_3$ m$^{-2}$ yr$^{-1}$ per % plant cover. We used these relationships to provide a basis for estimating carbonate production rates in Mar 2017 and Jan 2019 as a function of *Halimeda* spp. % cover. These carbonate production rates are equivalent to a rate of sediment generation because all calcified plant segments are ultimately shed into adjacent habitats [16]. Our estimated production rates for Mar 2017 and Sept 2019 are underpinned by two assumptions: (i) that the relative proportions of the two main *Halimeda* species have not changed significantly (an assumption supported by visual field observations in March 2017 and Jan 2019), and (ii) that rates of calcification and species turnover from the main *Halimeda* spp. have not changed over the past 2 years.

Benthic cover data, parrotfish biomass and calculated sediment production rates between subsequent sampling periods were tested for statistical significance using paired two-sided *t*-tests (electronic supplementary material, table S1).

## 2.3. Sediment grain-size contributions

To quantify the sedimentary contributions from both parrotfish bioerosion and *Halimeda* spp. calcification, i.e. the proportions of sediment in different grain-size classes being produced (following the Udden–Wentworth nomenclature), we used data from two sources. For parrotfish, we used previously collected data on the proportion of sediment produced within each standard grain-size class, as a function of body size class and life phase, for *Chlorurus sordidus*, *Ch. strongylocephalus*, *Scarus niger*, *S. frenatus*, *S. psittacus* and *S. rubroviolaceus* [29]. These data were then used to estimate the amount of sediment produced in each size class in each time period as a function of the calculated parrotfish sediment generation rate. For species without empirical grain-size data, we substituted data from the nearest equivalent sister species based on the taxonomy of Choat *et al.* [31]. For *Halimeda* spp., we used data on the sedimentary products, as calculated from plant breakdown experiments, for *H. macrophysa* and *H. micronesica* [30]. As with parrotfish, we then estimated the amount of sediment produced in each grain-size class as a function of the calculated production rate for each species in each survey period. All data are given as mean $\pm$ s.d.

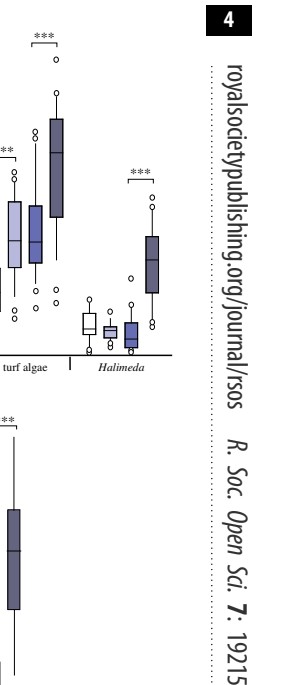

**Figure 1.** Box (median and 50% quartile) and whisker (95% quantile) plots (outlier points are outside the 95th percentile) showing changes across the four survey time periods (Jan 2016, Sept 2016, Mar 2017 and Jan 2019) in: (*a*) total parrotfish biomass (kg ha$^{-1}$); (*b*) parrotfish biomass (kg ha$^{-1}$) by species; (*c*) cover (%) of coral, turf/filamentous algae, and *Halimeda* spp.; (*d*) total parrotfish bioerosion/sediment generation (kg CaCO$_3$ m$^{-2}$ yr$^{-1}$); (*e*) parrotfish bioerosion/sediment generation (kg CaCO$_3$ m$^{-2}$ yr$^{-1}$) by species and (*f*) *Halimeda* spp. carbonate production (kg CaCO$_3$ m$^{-2}$ yr$^{-1}$). Data are pooled across sites within each time period. Significant differences within individual categories highlighted (*$p < 0.05$, **$p < 0.01$, ***$p < 0.001$). Parrotfish genera abbreviations in (*b*) and (*e*): *Ct.* – *Cetoscarus*; *Ch.* – *Chlorurus*; *S.* – *Scarus*; *H.* – *Hipposcarus*.

# 3. Results and discussion

## 3.1. Changes in parrotfish biomass and benthic community composition

Total parrotfish biomass (kg ha$^{-1}$) in the shallow reef habitat along the outer reef flat/upper reef slope increased significantly in the immediate aftermath of the mid-2016 coral bleaching event (Jan 2016: 126.7 ± 10.8 kg ha$^{-1}$; Sept 2016: 256.0 ± 42.2 kg ha$^{-1}$; $p < 0.001$) (figure 1*a*). Specifically, the biomass of all three common excavating parrotfish species (*Cetoscarus ocellatus*, *Chlorurus sordidus* and *Chlorurus strongylocephalus*) increased significantly over this period (*Ct. ocellatus* Jan 2016: 12.1 ± 10.9 kg ha$^{-1}$, Sept 2016: 28.9 ± 6.6 kg ha$^{-1}$, $p < 0.05$; *Ch. sordidus* Jan 2016: 53.5 ± 13.2 kg ha$^{-1}$, Sept 2016: 79.0 ± 17.7 kg ha$^{-1}$, $p < 0.05$; *Ch. strongylocephalus* Jan 2016: 19.5 ± 7.7 kg ha$^{-1}$; Sept 2016: 76.8 ± 4.1 kg ha$^{-1}$; $p < 0.01$) (figure 1*b*). Biomass of the scraper species *Scarus niger* and of *Hipposcarus harid* also increased significantly (*S. niger* Jan 2016: 18.4 ± 3.8 kg ha$^{-1}$, Sept 2016: 33.0 ± 13.2 kg ha$^{-1}$, $p < 0.05$; *H. harid* Jan 2016: 2.2 ± 3.5 kg ha$^{-1}$, Sept 2016: 12.4 ± 8.0 kg ha$^{-1}$, $p < 0.05$) (figure 1*b*). Conversely, biomass of other common *Scarus* species (*Scarus frenatus*, *Scarus psittacus* and *Scarus tricolor*) changed little over the immediate post-bleaching period (figure 1*b*). Total parrotfish biomass then remained relatively stable over the next approximately six months (to Mar 2017), but increased again between years 1 and 3 post-bleaching, equating to an overall approximately threefold increase by Jan 2019 (354.8 ± 124.5 kg ha$^{-1}$) (figure 1*a*). Of note, biomass of the excavator species *Ct. ocellatus* nearly doubled in this period (Mar 2017: 37.7 ± 24.2 kg ha$^{-1}$, Jan 2019: 68.9 ± 47.2 kg ha$^{-1}$; figure 1*b*).

These changes in parrotfish biomass occurred alongside a major restructuring of the reef benthic community (electronic supplementary material, figure S2). Bleaching severely impacted coral communities to depths of approximately 6–7 m, with widespread mortality of branching and digitate *Acropora* spp. [8]. As a result, live coral cover declined significantly in the immediate post-bleaching period (Jan 2016: 25.6 ± 8.0%, Sept 2016: 4.2 ± 2.7%, $p < 0.001$; figure 1*c*). Coral cover remained very low

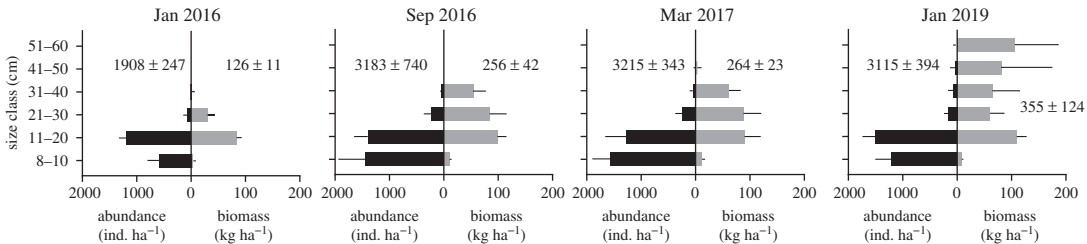

**Figure 2.** Parrotfish abundance (ind. ha$^{-1}$) and biomass (kg ha$^{-1}$) by size class (mean ± s.d.) in each of the four survey periods. Data are pooled across sites for each time period.

over the next approximately six months (Mar 2017: 4.7 ± 3.2%), before a slight increase over the subsequent 2 years (Jan 2019: 8.8 ± 4.7%) (figure 1c). Coral mortality in mid-2016 was accompanied by a significant increase in turf/filamentous algal cover, which rapidly colonized the still largely standing dead coral skeletons (Jan 2016: 14.7 ± 5.7%, Sept 2016: 25.7 ± 9.2%; figure 1c). Turf/filamentous algal cover then changed little between Sept 2016 and Mar 2017, but again increased significantly over the following 2 years (Mar 2017: 27.5 ± 10.3%, Jan 2019: 41.7 ± 13.8%; $p < 0.001$) (figure 1c). In the period from Mar 2017 to Jan 2019 (i.e. 1 to 3 years post-bleaching), we also observed a large increase in *Halimeda* spp. cover (figure 1c). *Halimeda* spp. was a relatively common benthic substrate colonizer pre-bleaching (Jan 2016: 6.4 ± 3.4%; figure 1c), often growing at the base of coral colonies and between dead coral branches (see [30]). *Halimeda* spp. cover changed little by Mar 2017, but increased significantly between Mar 2017 and Jan 2019 (Mar 2017: 4.9 ± 4.1%, Jan 2019: 20.5 ± 8.4%; $p < 0.001$) (figure 1c).

## 3.2. Implications for sediment generation rates

A major consequence of the increased biomass of parrotfish in the post-bleaching period has been a marked increase in rates of substrate bioerosion and sediment generation. Total rates of sediment generation by parrotfish increased significantly between Jan and Sept 2016 (Jan 2016: 0.5 ± 0.1 kg CaCO$_3$ m$^{-2}$ yr$^{-1}$, Sept 2016: 1.9 ± 0.5 kg CaCO$_3$ m$^{-2}$ yr$^{-1}$; $p < 0.001$) (figure 1d). This approximately fourfold increase occurred mainly as a function of the increased biomass of excavating species, with particularly large increases for both *Ct. ocellatus* ($p < 0.01$) and *Ch. strongylocephalus* ($p < 0.001$) (figure 1e). Smaller increases in biomass were observed for some scraper species (figure 1b); however, this equated to little or no increase in sediment generation (figure 1e). Total parrotfish sediment generation remained relatively constant through Mar 2017, but had increased by a further approximately 50% by Jan 2019 (Mar 2017: 2.1 ± 0.5 kg CaCO$_3$ m$^{-2}$ yr$^{-1}$, Jan 2019: 3.5 ± 1.9 kg CaCO$_3$ m$^{-2}$ yr$^{-1}$; figure 1d). The majority of this increase can be attributed to further increases in biomass of *Ct. ocellatus*, doubling the bioerosion rates of this species (Mar 2017: 0.5 ± 0.4 kg CaCO$_3$ m$^{-2}$ yr$^{-1}$, Jan 2019: 1.3 ± 0.9 kg CaCO$_3$ m$^{-2}$ yr$^{-1}$) (figure 1e). Scraper species continued to make little contribution to overall bioerosion rates throughout this period.

We consider the most likely explanation for such increases in bioerosion and sediment generation rates by parrotfish to be an increase in food resource availability as turf algae colonized dead coral substrates following the bleaching event. Parrotfish are microphages that target epi- and endolithic cyanobacteria and chlorophytes as their primary food source [32]. Such food sources are naturally present within both living coral (often as an *Ostreobium* spp. band) and both within and on dead coral substrates (the latter being referred to as turf assemblages). Recent evidence shows that extensive biofilms and rapid expansion of epi- and endolithic cyanobacteria and chlorophyte communities occur almost immediately after bleaching-induced coral mortality [33]. This makes available a huge food resource for parrotfish exploitation and, as recent work has shown, this is a likely driver of immediate post-bleaching increases in parrotfish growth rates [34]. We suggest that the magnitude of this increase has been especially marked at these Maldivian sites because of the high (pre-bleaching) cover of complex branching *Acropora* spp. Conversely, reefs with existing low structural complexity may not have experienced such significant increases. We have no data on the composition of the phototropic microbial and filamentous algal communities that bloomed post-bleaching in the Maldives, but the rapid expansion of these communities that occurred on dead standing *Acropora* (and other dead coral) substrate immediately after bleaching at these sites (see electronic supplementary material, figure S3) suggests that a major increase in food resources occurred.

This inferred response by parrotfish to food availability is supported by an analysis of changes in parrotfish abundance (ind. ha$^{-1}$) and biomass (kg ha$^{-1}$) throughout the 3-year period of our study (figure 2). This shows: (i) that the abundance of small parrotfish (≤ 20 cm) increased markedly in the first

few months post-bleaching—both as a function of food availability but also potentially new recruitment; (ii) that this was accompanied by an increase in biomass in these same size classes; but also (iii) that the major proportion of total biomass increase was due to smaller increases in the abundance of medium-sized parrotfish (21–40 cm) between Jan and Sept 2016, and of large parrotfish (41–60 cm) between Mar 2017 and Jan 2019 (figure 2). The increased biomass observed in this 1- to 3-year period post-bleaching has thus mainly been associated with increasing fish size rather than overall increases in the abundance of parrotfish (figure 2), a response consistent with increasing food resource availability. We note that such rapid rates of growth, especially for juvenile fish, are well within the rates reported for parrotfish (which can grow to in excess of 20 cm within 2 years [35]), and that these findings are consistent with empirical data from sites in Chagos and on the Great Barrier Reef that show increasing rates of parrotfish growth immediately after the 2016 bleaching [34]. We also note that the most significant biomass (figure 1$b$) and abundance increases over the 3-year period occurred in the excavator species (excavators: Jan 2016: 892 ± 83 ind. ha$^{-1}$, Jan 2019: 1823 ± 378 ind. ha$^{-1}$; scrapers: Jan 2016: 1016 ± 209 ind. ha$^{-1}$, Jan 2019: 1292 ± 501 ind. ha$^{-1}$). We suggest that this is consistent with the capacity of these species to more effectively exploit the rapidly expanding endolithic communities that develop because of their more aggressive (excavating) feeding mode [21].

Alongside the increased sediment production from parrotfish, we also observe an approximately threefold increase in *Halimeda* spp. sediment production associated with a later stage of the post-bleaching period. *Halimeda* sediment production was estimated to be approximately 0.07 kg CaCO$_3$ m$^{-2}$ yr$^{-1}$, in Jan 2016 when *Halimeda* spp. cover was approximately 6% and approximately 0.06 kg CaCO$_3$ m$^{-2}$ yr$^{-1}$ in Sept 2016 when *Halimeda* spp. cover was approximately 5%. Estimated production stayed constant for the following six months and then increased significantly between Mar 2017 and Jan 2019 (Mar 2017; 0.05 ± 0.02 kg CaCO$_3$ m$^{-2}$ yr$^{-1}$, Jan 2019; 0.23 ± 0.04 kg CaCO$_3$ m$^{-2}$ yr$^{-1}$; $p < 0.001$) (figure 1$e$). This occurred as a function of the rapid expansion of *Halimeda* spp. cover, especially beneath and between the branches of dead branching *Acropora* spp., which created abundant semi-cryptic habitat space protected from predation (electronic supplementary material, figure S4). We note that a similar rapid expansion of *Halimeda* spp. cover has occurred on other post-bleaching reefs in the central Indian Ocean [36]. The resulting effect on sediment production increase, however, is almost a magnitude smaller than that caused by parrotfish. However, we consider our estimates are probably conservative, because they are based only on % cover of *Halimeda* spp. when in reality the overall biomass of *Halimeda* spp has also increased significantly.

## 3.3. Changes in sediment grain-size production

An additional question that arises as a function of increased overall sediment generation rates is what implications have these changes had on the amounts of sediment being produced within different grain-size fraction classes. This is an important question because the shorelines and islands that reefs supply with sediment not only typically comprise a narrow suite of sediment types derived from reef organisms/processes, but also comprise a restricted range of sediment grain-size classes—as indeed is the case for the islands adjacent to these reefs [30]. In this context, it would be reasonable to hypothesize that the major increase in biomass of excavating species of parrotfish that has occurred at our sites would have markedly changed not only total rates of sediment generation, but also the proportions of sediment produced in different size classes. However, our data from across the four surveys periods (figure 3) show that although the total amount of carbonate being generated in each grain-size class has indeed increased, the unimodal size class distribution of sediments has remained broadly consistent throughout (figure 3). This is a function of the very consistent grain sizes of sediment that are generated by different parrotfish species in this region regardless of species, body size class and feeding mode [29]. Thus, the net impact has been to increase sediment production of all grain-size classes, but without major changes in the proportions of sediment sizes generated. While our study focused only on sediment production increases by parrotfish and *Halimeda* spp. (because past data show these contributors dominate Maldivian island sediments [10,37]), it is possible that other producers may also increase sediment supply post-bleaching. One specific possibility is that rates of sediment generation by endolithic sponges, which can aggressively colonize dead reefal substrate, may now also be producing more sediment, although if this is the case it is worth noting that these sponges mainly produce silt and mud grade carbonate that is not a major island contributor. However, overall the large rates of production that derive from parrotfish suggest they are likely to have the predominant influence on resultant patterns of sediment generation.

## 3.4. Conclusions and wider implications

Data presented in this paper clearly demonstrate that in addition to fundamentally changing coral community composition and reef-associated species abundance, thermally driven coral mortality can lead

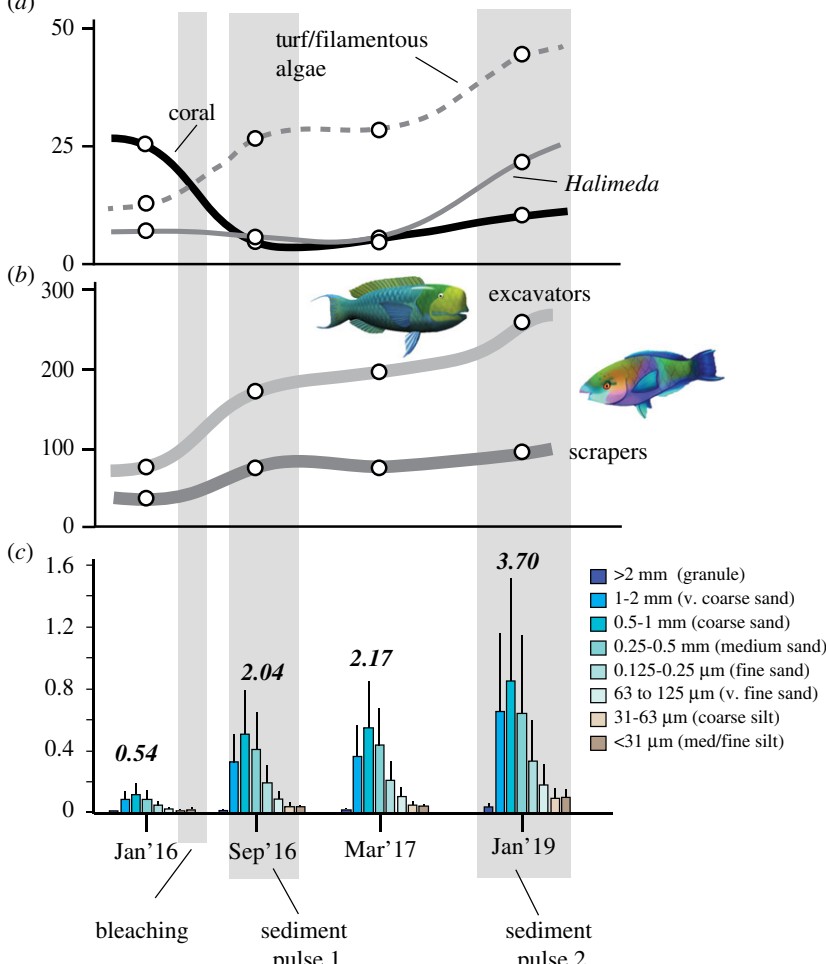

**Figure 3.** Summary diagram showing trends in (*a*) mean % coral, turf/filamentous algal and *Halimeda* spp. cover and (*b*) biomass (kg ha$^{-1}$) of excavator and scraper parrotfish species over time. White circles are mean values for each time period for each category. (*c*) Total calculated sediment generation rate (kg CaCO$_3$ m$^{-2}$ yr$^{-1}$) by parrotfish and *Halimeda* spp. combined (bold italics) and histograms showing the proportions (mean ± s.d.) of total sediment production by grain-size class. Grey bars show the timing of the bleaching event, and the two identified pulses of increased sediment generation.

to major changes in sediment yields from reefs. Different disturbances are likely to change reef ecology in different ways, and with different sediment production outcomes, but our data show that the post-bleaching period at these southern Maldives sites has been defined by rapid changes in the abundance of two locally important sediment producing taxa (parrotfish and *Halimeda* spp.). These changes are not a direct function of the warming event itself, but rather have been facilitated by the benthic ecological transitions that have occurred as a result of the bleaching mortality. The net effect has been a major increase in reef-derived sediment production, with parrotfish and *Halimeda* spp. collectively contributing an additional approximately 3 kg sedimentary CaCO$_3$ m$^{-2}$ yr$^{-1}$ by 3 years post-bleaching (increasing from an estimated approximately 0.5 to approximately 3.7 kg CaCO$_3$ m$^{-2}$ yr$^{-1}$). This overall increase in sediment production has occurred through two distinct phases or pulses of enhanced sediment generation (figure 3). The first occurred soon after bleaching (by five to six months post-bleaching) as parrotfish abundance and biomass increased, which resulted in higher rates of parrotfish bioerosion and sediment production. The second sediment generation pulse occurred between 1 and 3 years post-bleaching with a further increase in parrotfish biomass and a major (approx. fourfold) increase in *Halimeda* spp. abundance. Such changes strongly support existing conceptual ideas about the way in which sediment generation rates may respond to ecological transitions, either driven by natural or disturbance-driven [18] events, but here we provide novel data on the magnitude and timing of these changes.

These changes not only demonstrate the wider carbonate budget implications of coral bleaching, but also have potentially important implications for adjacent reef landforms within atoll reef settings that are solely sustained by reef-derived sediment supply, in this case reef island shorelines and lagoons. Most relevant

from an island/shoreline maintenance perspective has been the large increase, over a very short time scale, in the amount of fine-grained to very coarse-grained sand being produced. This is important because most of the sediments within the atoll interior islands in this area of the Maldives also comprise mainly medium- to coarse-grained sands produced by corals and coralline algae (approx. 60–70%) [30]. If this largely parrotfish-derived material (as it seems reasonable to suggest) has similar transport potential to the existing sediment pool, this could significantly increase (by a factor of approx. 3) sediment supply to the islands, potentially driving increases in sediment accumulation along the island shoreface.

Our data also show that sediment generation rates by *Halimeda* spp. increased markedly in the 3 years since bleaching, although the magnitude of sediment generated compared with parrotfish is much smaller. Nonetheless, this still represents an important additional sedimentary input. The fate of this material in terms of its potential to contribute to the proximal islands is less clear. This is because: (i) past experimental work has shown that the *Halimeda* spp. that dominate at these sites (*H. micronesica* and *H. macrophysa*) rapidly break down to fine sand- and silt-sized fractions (which are minor components of the island sediments), and (ii) that disarticulated whole *Halimeda* plates are only a very minor island sediment constituent at these sites [30]. Thus, while *Halimeda* segments are abundant within sediments on the outer reef flat and the upper fore-reef and may be preservable as discrete event horizons within the reef framework, this material may not, in contrast with the parrotfish-derived material, contribute as much to increasing island sediment supply due to higher post-depositional breakdown rates.

What is less clear are the time scales over which these higher rate production (and supply) scenarios may persist, but it is reasonable to assume that this may be a relatively short-lived pulse event. As the structure of the now dead reef community breaks down, a necessary prelude to coral recolonization, so both the substrate that hosts abundant food resources for the parrotfish and the cryptic habitat space for *Halimeda* colonization may progressively denude. This may occur over the next 4–5 years (i.e. by 7 to 8 years post-bleaching) assuming similar time scales of framework denudation as followed the 1998 bleaching event in this region [38]. A caveat here is that this may also depend on rates of reef recovery. Robinson *et al.* [6] have, for example, shown persistent shifts in the abundance of herbivorous taxa (including parrotfish) on phase-shifted reefs in the Seychelles, and future recovery trajectories and/or timescales of bleaching recurrence that may reset any recovery phase may further influence sediment generation regimes. These ideas set up some interesting hypotheses about the feedbacks that may occur between reef structure, reef 'health' and the abundance of reef-associated sediment-generating taxa, and thus the potentially cyclical nature of phases of higher and lower rates of sediment generation associated with major disturbance events.

Ethics. Permission to undertake fieldwork was granted to C.T.P. by the Ministry of Fisheries, Marine Resources and Agriculture, Male, Maldives (permit nos. 30-D/INDIV/2015/451 and 30-D/INDIV/2018/1128). The study did not involve the handling or manipulation of animal subjects or tissues, and thus no additional ethical permissions were needed.

Data accessibility. Benthic ecological data, and parrotfish abundance and biomass data used in this study are deposited at the Dryad Digital Repository: https://doi.org/10.5061/dryad.08kprr4zc [39].

Authors' contributions. C.T.P. conceived the study and wrote the first draft. C.T.P. and K.M.M collected the field data. All authors contributed to data interpretation, revised the manuscript and approved the submitted version.

Competing interests. The authors declare no competing interests.

Funding. Research was supported through a Leverhulme Research Fellowship (grant no. RF-2015-152) to C.T.P., with additional support provided by the Bertarelli Foundation through a Bertarelli Program in Marine Science award to C.T.P.

Acknowledgements. We thank Mohamed Aslam and staff at the LaMer Small Island Research Centre, Faaresmathooda for their invaluable help and assistance with fieldwork and logistics. Two anonymous reviewers are thanked for their helpful comments.

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
