## [Reviewer comments · Royal Society Open Science]

Review History

RSOS-192153.R0 (Original submission)

Review form: Reviewer 1

Is the manuscript scientifically sound in its present form?

No

Are the interpretations and conclusions justified by the results?

Yes

Is the language acceptable?

Yes

Do you have any ethical concerns with this paper?

No

Have you any concerns about statistical analyses in this paper?

No

Recommendation?

Accept with minor revision (please list in comments)

Comments to the Author(s)

Review of Perry et al., "Bleaching-driven reef community shifts..."

This MS describes changes in rates and types of sediment production from bleached reefs in Maldives. They find two pulses of sediment production: one, a few months after bleaching, which they ascribe to parrotfish jumping all over the new substrate with its algal cohorts, and the other, two years later, from Halimeda increases.

As it stands, the paper is OK-acceptable, a useful addition to our store of knowledge about this vanishing ecosystem. I have some minor quibbles, and would like some major holes filled.

First of all: this is minor, but one of my pet peeves. We know, from the Numbers Theory we learned in Year 1, that experimental data may only be reported to one figure less than the measurement precision. We know that in theory, but somehow can't resist the siren call of those extra numbers...hence, we get typical field techniques-that somehow report sediment production rates to 4 figures, with a precision of +/- 5g. This is unrealistic. Chop a few figures off, here-no one will care, and you will feel better about the whole thing.

I question the "bioerosion" part of this...previous work in Maldives (Hussain Zahir, for example) showed the usual suite of sponges, worms and bivalves doing great damage. Other authors have documented increases in internal bioerosion following bleaching events-yet this MS reads as though only 2 groups of organisms are involved. For example, others have shown that boring sponges produce more sand-sized material than grazers. I suggest authors at least mention in passing that they are looking at only part of the picture.

I am also intrigued by the Discussion...Perry's previous work has shown that the carbonate budget of Maldivian reefs is negative, or can be...calcn rates of 5kg/m²/yr. The MS reports that parrotfish grazing went from a few 100g to 5kg. The Discussion seems to centre around making beaches for resorts, but it seems to me it should say, get those resorts the hell outta there because the reefs are disappearing.

I am reminded of some work I did yonks ago, on a Costa Rican reef hit by the 1988 El Nino bleaching. Corals all dead, framework being chewed up...reef disappearing. So I thought, Phantom Reefs. Future geologists will never see those Costa Rican reefs, never know they were there...except for a layer of reef-derived sediment grains in the offshore.

I made a shockingly deep dive, retrieved a core with a lovely bone-white layer of framework-derived sediment, got attacked by a shark, lived to write an Abstract which damned if I can find now.

Review form: Reviewer 2

Is the manuscript scientifically sound in its present form?

Yes

Are the interpretations and conclusions justified by the results?

Yes

Is the language acceptable?

Yes

Do you have any ethical concerns with this paper?

No

Have you any concerns about statistical analyses in this paper?

No

Recommendation?

Accept as is

Comments to the Author(s)

A paper by Perry et al. presented an interesting case study that ecological changes link to geological/sedimentary process by demonstrating that coral bleaching induces increasing sediment generation by bioeroders (parrotfish) and calcareous green algae at Maldivian reefs following the 2016 bleaching event. This paper would be of interest to wider audiences studying biological-geological interactions in coral reef and other ecosystems, and stimulate similar studies in other reef regions. I recommend that the paper is acceptable as is, but I feel it little bit lengthy; it could be compacted. I have only few comments on the manuscript.

The main comment is about the magnitude and timescales for these ecological changes to be seen in geological/sedimentary changes, as discussed in the last section by the authors. As authors showed increasing sediment generation but a consistent sediment size distribution during the study period, these two-pulsed increases in sediment may not be preserved in sedimentary records. I am wondering what extent of ecological changes can be detected in the sedimentary records. I am also wondering if these ecological changes will reflect obvious changes in shoreline and sediment volumes. If not, how long more years are necessary to reflect in shoreline and sediment volume changes? Does it take decade, hundred, millennium? I look forward to Authors' future works on longer-term monitoring in sediment generation and volume, to answer these questions and demonstrate changes in shorelines and island topography.

Another comment is about the recruitment of parrotfish. I generally agree with authors' conclusion that increasing parrotfish abundance and biomass is caused by an increase in food availability following the 2016 bleaching event. However, as authors noted, increasing abundance in small parrotfish populations needs new recruitment. I am not sure about the life history of parrotfishes, but I am wondering how long it takes for parrotfish to get 8-10 cm size class. When is their main breeding season? Increasing small parrotfish abundance following the 2016 bleaching event is maybe due to the loss of predators for higher temperature or other reasons. I suggest that authors could provide basic biological information on parrotfish to support their conclusion.

Although some snapshot underwater images are available in the ESM files, I would like to see visual images showing benthic cover changes during the study period.

Decision letter (RSOS-192153.R0)

27-Feb-2020

Dear Dr Perry

On behalf of the Editors, I am pleased to inform you that your Manuscript RSOS-192153 entitled "Bleaching-driven reef community shifts drive pulses of increased reef sediment generation" has been accepted for publication in Royal Society Open Science subject to minor revision in

accordance with the referee suggestions. Please find the referees' comments at the end of this email.

The reviewers and handling editors have recommended publication, but also suggest some minor revisions to your manuscript. Therefore, I invite you to respond to the comments and revise your manuscript.

- Ethics statement

- Data accessibility

If you wish to submit your supporting data or code to Dryad (<http://datadryad.org/>), or modify your current submission to dryad, please use the following link:
<http://datadryad.org/submit?journalID=RSOS&manu=RSOS-192153>

- Competing interests

- Authors' contributions

- Acknowledgements

- Funding statement

Because the schedule for publication is very tight, it is a condition of publication that you submit the revised version of your manuscript before 07-Mar-2020. Please note that the revision deadline will expire at 00.00am on this date. If you do not think you will be able to meet this date please let me know immediately.

Please note that Royal Society Open Science charge article processing charges for all new submissions that are accepted for publication. Charges will also apply to papers transferred to

Royal Society Open Science from other Royal Society Publishing journals, as well as papers submitted as part of our collaboration with the Royal Society of Chemistry (<https://royalsocietypublishing.org/rsos/chemistry>).

If your manuscript is newly submitted and subsequently accepted for publication, you will be asked to pay the article processing charge, unless you request a waiver and this is approved by Royal Society Publishing. You can find out more about the charges at <https://royalsocietypublishing.org/rsos/charges>. Should you have any queries, please contact openscience@royalsociety.org.

Kind regards,

Anita Kristiansen
Editorial Coordinator

on behalf of Jon Blundy (Subject Editor)
openscience@royalsociety.org

Associate Editor Comments to Author:

Comments to the Author:

Thank you for transferring this manuscript to Royal Society Open Science, we are grateful for the support. Following review by two referees, it appears that, subject to revisions, your manuscript may be accepted for publication. Please carefully address the referees' commentary in your revised manuscript, as well as response to reviewers that delineates the changes made, and your responses to the reviewer commentaries.

Reviewer comments to Author:

Reviewer: 1

Comments to the Author(s)

Review of Perry et al., "Bleaching-driven reef community shifts..."

This MS describes changes in rates and types of sediment production from bleached reefs in Maldives. They find two pulses of sediment production: one, a few months after bleaching, which they ascribe to parrotfish jumping all over the new substrate with its algal cohorts, and the other, two years later, from Halimeda increases.

As it stands, the paper is OK-acceptable, a useful addition to our store of knowledge about this vanishing ecosystem. I have some minor quibbles, and would like some major holes filled.

First of all: this is minor, but one of my pet peeves. We know, from the Numbers Theory we learned in Year 1, that experimental data may only be reported to one figure less than the measurement precision. We know that in theory, but somehow can't resist the siren call of those extra numbers...hence, we get typical field techniques-that somehow report sediment production rates to 4 figures, with a precision of +/- 5g. This is unrealistic. Chop a few figures off, here-no one will care, and you will feel better about the whole thing.

I question the “bioerosion” part of this...previous work in Maldives (Hussain Zahir, for example) showed the usual suite of sponges, worms and bivalves doing great damage. Other authors have documented increases in internal bioerosion following bleaching events-yet this MS reads as though only 2 groups of organisms are involved. For example, others have shown that boring sponges produce more sand-sized material than grazers. I suggest authors at least mention in passing that they are looking at only part of the picture.

I am also intrigued by the Discussion...Perry’s previous work has shown that the carbonate budget of Maldivian reefs is negative, or can be...calcn rates of 5kg/m²/yr. The MS reports that parrotfish grazing went from a few 100g to 5kg. The Discussion seems to centre around making beaches for resorts, but it seems to me it should say, get those resorts the hell outta there because the reefs are disappearing.

I am reminded of some work I did yonks ago, on a Costa Rican reef hit by the 1988 El Nino bleaching. Corals all dead, framework being chewed up...reef disappearing. So I thought, Phantom Reefs. Future geologists will never see those Costa Rican reefs, never know they were there...except for a layer of reef-derived sediment grains in the offshore.

I made a shockingly deep dive, retrieved a core with a lovely bone-white layer of framework-derived sediment, got attacked by a shark, lived to write an Abstract which damned if I can find now.

Reviewer: 2

Comments to the Author(s)

A paper by Perry et al. presented an interesting case study that ecological changes link to geological/sedimentary process by demonstrating that coral bleaching induces increasing sediment generation by bioeroders (parrotfish) and calcareous green algae at Maldivian reefs following the 2016 bleaching event. This paper would be of interest to wider audiences studying biological-geological interactions in coral reef and other ecosystems, and stimulate similar studies in other reef regions. I recommend that the paper is acceptable as is, but I feel it little bit lengthy; it could be compacted. I have only few comments on the manuscript.

The main comment is about the magnitude and timescales for these ecological changes to be seen in geological/sedimentary changes, as discussed in the last section by the authors. As authors showed increasing sediment generation but a consistent sediment size distribution during the study period, these two-pulsed increases in sediment may not be preserved in sedimentary records. I am wondering what extent of ecological changes can be detected in the sedimentary records. I am also wondering if these ecological changes will reflect obvious changes in shoreline and sediment volumes. If not, how long more years are necessary to reflect in shoreline and sediment volume changes? Does it take decade, hundred, millennium? I look forward to Authors’ future works on longer-term monitoring in sediment generation and volume, to answer these questions and demonstrate changes in shorelines and island topography.

Another comment is about the recruitment of parrotfish. I generally agree with authors’ conclusion that increasing parrotfish abundance and biomass is caused by an increase in food availability following the 2016 bleaching event. However, as authors noted, increasing abundance in small parrotfish populations needs new recruitment. I am not sure about the life history of parrotfishes, but I am wondering how long it takes for parrotfish to get 8-10 cm size class. When is their main breeding season? Increasing small parrotfish abundance following the 2016 bleaching event is maybe due to the loss of predators for higher temperature or other reasons. I suggest that authors could provide basic biological information on parrotfish to support their conclusion.

Although some snapshot underwater images are available in the ESM files, I would like to see visual images showing benthic cover changes during the study period.

Author's Response to Decision Letter for (RSOS-192153.R0)

See Appendix A.

Decision letter (RSOS-192153.R1)

27-Mar-2020

Dear Dr Perry,

It is a pleasure to accept your manuscript entitled "Bleaching-driven reef community shifts drive pulses of increased reef sediment generation" in its current form for publication in Royal Society Open Science.

Best regards,

on behalf of the Associate Editor, and Professor Jon Blundy (Subject Editor)
openscience@royalsociety.org

Appendix A

Manuscript ID RSOS-192153

Bleaching-driven reef community shifts drive pulses of increased reef sediment generation

Chris T. Perry; Kyle M. Morgan, Ines D. Lange, and Robert T. Yarlett

We thank both reviewers for their input and comments on the manuscript, which we have responded to as outlined below.

Responses to Reviewer: 1

This MS describes changes in rates and types of sediment production from bleached reefs in Maldives. They find two pulses of sediment production: one, a few months after bleaching, which they ascribe to parrotfish jumping all over the new substrate with its algal cohorts, and the other, two years later, from *Halimeda* increases.

Comment 1.1 First of all: this is minor, but one of my pet peeves. We know, from the Numbers Theory we learned in Year 1, that experimental data may only be reported to one figure less than the measurement precision. We know that in theory, but somehow can't resist the siren call of those extra numbers...hence, we get typical field techniques-that somehow report sediment production rates to 4 figures, with a precision of +/- 5g. This is unrealistic. Chop a few figures off, here-no one will care, and you will feel better about the whole thing.

Response: We have only reported data to two decimal places in the text, but in light of the reviewers comments these have been rounded to 1 decimal place for parrotfish production estimates. For Halimeda we are dealing with very light weights for even whole plants and thus even large increases in plant numbers may result in only a few 10's to 100 g of additional carbonate being produced. In this case, because we report changes in kg to be consistent with the parrotfish metrics, working at 2 decimal places seems appropriate.

Comment 1.2: I question the "bioerosion" part of this...previous work in Maldives (Hussain Zahir, for example) showed the usual suite of sponges, worms and bivalves doing great damage. Other authors have documented increases in internal bioerosion following bleaching events-yet this MS reads as though only 2 groups of organisms are involved. For example, others have shown that boring sponges produce more sand-sized material than grazers. I suggest authors at least mention in passing that they are looking at only part of the picture.

Response: The reviewer is entirely correct that other bioeroding taxa will also contribute to substrate erosion post-bleaching, specifically internal bioeroders. We are not discounting their additive role here, but there exist no viable methods for measuring changes in internal bioerosion rates based on census data, and we have no pre-event data to compare this against. However, we have now added a brief comment into the discussion emphasising that the activities of these internal eroders may be further exacerbating rates of substrate erosion and sediment supply (see tracked changed version – lines 297 onwards). However, this would only derive from sponges – worms and bivalves excavate chemically through dissolution and do not produce sediment. Sponges do produce sediment, but this is all in the fine sand fraction and (mostly below this) size fractions i.e., silt/mud grade, as recent work by, for example, deBakker has shown.

Comment 1.3: I am also intrigued by the Discussion...Perry's previous work has shown that the carbonate budget of Maldivian reefs is negative, or can be...calcn rates of 5kg/m²/yr. The MS reports that parrotfish grazing went from a few 100g to 5kg. The Discussion seems to centre around making beaches for resorts, but it seems to me it should say, get those resorts the hell outta there because the reefs are disappearing.

Response: Here we assume that the reviewer is referring to a paper we wrote in 2017 about the immediate post-bleaching change in predicted reef budget states, which showed a transition over the first 6 months - shifting from strongly net positive (mean 5.92 G, where G = kg CaCO₃ m⁻² yr⁻¹) to strongly net negative (mean -2.96 G). This early transition was driven both by widespread coral mortality, but also an estimated increase in parrotfish erosion. This is the same early increase in parrotfish erosion we report here, but which we now (here) explore over the subsequent 3 years through two further surveying trips. We also explore here for the first time the impacts on sediment generation – this is now possible because of the sediment production grain size data we now have available (as we report in the paper).

In terms of sediment supply to beaches – we do not have any empirical data on the actual transfer rates of sediment from reef to island, so it is hard to evaluate precisely the impact on the transport pathways. One might assume a short-term spike, but that this may decline as the reef denudes. However, as the reviewer goes on to point out in his Costa Rican example – it is also possible that much of this sediment may end up off-reef – again our ability to predict that is limited and really the point of this paper was to provide some numbers on how sediment generation rates can change post-bleaching.

Response to Reviewer: 2

Comments to the Author(s)

A paper by Perry et al. presented an interesting case study that ecological changes link to geological/sedimentary process by demonstrating that coral bleaching induces increasing sediment generation by bioeroders (parrotfish) and calcareous green algae at Maldivian reefs following the 2016 bleaching event. This paper would be of interest to wider audiences studying biological-geological interactions in coral reef and other ecosystems, and stimulate similar studies in other reef regions. I recommend that the paper is acceptable as is, but I feel it little bit lengthy; it could be compacted. I have only few comments on the manuscript.

Response. Thanks for the supportive comments. We have tried to edit the text a little further but were also struggling to see where major changes could be made

Comment 2.1 The main comment is about the magnitude and timescales for these ecological changes to be seen in geological/sedimentary changes, as discussed in the last section by the authors. As authors showed increasing sediment generation but a consistent sediment size distribution during the study period, these two-pulsed increases in sediment may not be preserved in sedimentary records. I am wondering what extent of ecological changes can be detected in the sedimentary records. I am also wondering if these ecological changes will reflect obvious changes in shoreline and sediment volumes. If not, how long more years are necessary to reflect in shoreline and sediment volume changes? Does it take decade, hundred, millennium? I look forward to Authors' future works on longer-term monitoring in sediment generation and volume, to answer these questions and demonstrate changes in shorelines and island topography.

Response: the question of how long it may take to see a quantifiable shift in island shorelines is very interesting and something we have discussed amongst ourselves (and indeed the lead author has had discussions over this question with those working on remote sensing change assessments). The short and simple answer is that we do not know. I am aware of attempts to look at imagery from 2 years post bleaching in the Chagos that cannot yet, at the resolution of the available imagery, discern change that can be definitely ascribed to bleaching – i.e., not outside the expected inter-annual range. Clearly, as we move deeper into the post-bleaching period such changes in island shorelines may become visible – and, as above I am aware of remote sensing groups looking at just these issues in other areas. The other point that follows here is that given the inherent sediment mixing (wave drive and bioturbation driven) that occurs along the margins of these islands there is no obvious evidence of a clear sedimentary layer from this change – at least not yet. However, the fore-reef sediments are rich in Halimeda and one might imagine this would form a discrete horizon if preserved in the interstitial cavities in the reef framework (see comment in tracked changed version – line 349)

Comment 2.2 Another comment is about the recruitment of parrotfish. I generally agree with authors' conclusion that increasing parrotfish abundance and biomass is caused by an increase in food availability following the 2016 bleaching event. However, as authors noted, increasing abundance in small parrotfish populations needs new recruitment. I am not sure about the life history of parrotfishes, but I am wondering how long it takes for parrotfish to get 8-10 cm size class. When is their main breeding season? Increasing small parrotfish abundance following the 2016 bleaching event is maybe due to the loss of predators for higher temperature or other reasons. I suggest that authors could provide basic biological information on parrotfish to support their conclusion.

Response: Thanks for the interesting comments here. One key paper to mention here is the work of Taylor et al. (2019) which we cite, that has also shown rapid growth and increased population size in parrotfish after bleaching (see our discussion). As that paper states "Our review of the literature suggests that parrotfishes typically respond to disturbance by increasing in numbers with a peak occurring approximately 2 years after the event. Numerical densities at this time are a factor of two to eight times the pre-disturbance densities ". These patterns basically fit the trends we are seeing here. We can find no data on peak recruitment times for parrotfish in this region but, in general, most species tend to spawn year round, often on lunar cycles. Hence, any cohort has a wide range of ages across the year because they don't have a single period of mass spawning and subsequent recruitment. It is possible that the increased food quality could simply have led to higher survival rates/growth rates for juveniles that are constantly present. Of direct relevance here to note is that MSc thesis data (Barba, Jacquelyn (2010) Demography of parrotfish: age, size and reproductive variables. Masters (Research) thesis, James Cook University) and data published by Taylor et al. 2018 (which we now cite) shows that parrotfish can grow to 10 cm in the first 6-9 months of their lives, so the size increases we observe are not unexpected in this context. We have added a line or two on this topic to the discussion (see tracked changed version, line 251 on).

Comment 2.3 Although some snapshot underwater images are available in the ESM files, I would like to see visual images showing benthic cover changes during the study period.

Response: We have added an additional ESM (ESM fig 2) showing images from each of the four sampling periods taken as roughly the same position on one of the reefs. We hope you find this informative.